# Subglacial discharge effects on basal melting of a rotating, idealized ice shelf

Irena Vaňková[1], Xylar Asay-Davis[1], Carolyn Branecky Begeman[1], Darin Comeau[1], Alexander Hager[1], Matthew Hoffman[1], Stephen F. Price[1], and Jonathan Wolfe[1]

[1]Los Alamos National Laboratory, Los Alamos, NM 87545, USA

**Correspondence:** Irena Vaňková (vankova@lanl.gov)

**Abstract.** When subglacial meltwater is discharged into the ocean at the grounding line, it acts as a source of buoyancy, enhancing flow speeds along the ice base that result in higher basal melt rates. The effects of subglacial discharge have been well studied in the context of a Greenland-like, vertical calving front, where Earth's rotation can be neglected. Here we study these effects in the context of Antarctic ice shelves, where rotation is important. We use a numerical model to simulate ocean circulation and basal melting beneath an idealized three-dimensional ice shelf and vary the rate and distribution of subglacial discharge. For channelized discharge, we find that in the rotating case total melt-flux anomaly increases with two-thirds power of the discharge, in contrast with existing non-rotating results for which the melt-flux anomaly increases with one-third power of the discharge. The higher melt-flux anomaly with discharge is attributed to a more extensive area of the ice-shelf base being exposed to direct high melting by the rising plume as it is deflected due to Earth's rotation and its path is prolonged. For distributed discharge, we find that in both rotating and non-rotating cases the melt-flux anomaly increases with two-thirds power of the discharge. Furthermore, in the rotating case, the addition of channelized subglacial discharge can produce either higher or lower ice-shelf basal melt-flux anomaly than the equivalent amount of distributed discharge, depending on its location along the grounding line relative to the directionality of the Coriolis force. This contrasts with previous results from non-rotating, vertical ice-cliff simulations, where distributed discharge was always found to be more efficient at enhancing terminus-averaged melt rate than channelized discharge. The implication, based on our idealized simulations, is that melt-rate parameterizations attempting to include subglacial discharge effects that are not geometry and rotation aware may produce total melt-flux anomalies that are off by a factor of two or more.

## 1   Introduction

Subglacial freshwater originates as meltwater formed at the glacier bed, or as meltwater draining there from supraglacial or englacial sources. It is discharged into the ocean at the grounding line, where the ice goes afloat. Being released at depth, this freshwater is a source of buoyancy that accelerates as an ascending meltwater plume along the ice-shelf base or the vertical calving face, enhancing entrainment of ambient waters, and modulating melt rates. In Alaska and Greenland, where supraglacial melting is a significant contributor to subglacial discharge, local measurements near marine-terminating glaciers have shown such high seasonal submarine melting that it could only have been driven by enhanced subglacial discharge from

summer surface melt (Motyka et al., 2003; Washam et al., 2019). Circulation and renewal in some fjords has been shown to be driven almost entirely by subglacial discharge (Gladish et al., 2015; Carroll et al., 2017; Slater et al., 2018; Hager et al., 2022b). Meanwhile, the importance of subglacial discharge on sub-ice shelf circulation and melting in Antarctica is less clear, because the drainage of supraglacial meltwater to the bed is lacking, and present-day subglacial discharge is sourced only from much smaller magnitude subglacial melting. However, there has been recent surge of interest in the influence of subglacial discharge on basal melting beneath Antarctic ice shelves. Dow et al. (2022) and Hager et al. (2022a) demonstrate that the large catchment sizes in some parts of Antarctica, when combined with high subglacial melting, can lead to subglacial discharge fluxes similar in magnitude to the surface-derived melt fluxes in Greenland. Dow et al. (2022) found correspondence between elevated satellite-derived basal melt rates near grounding lines and locations of modeled subglacial meltwater discharge at some ice shelves. Pelle et al. (2023) found that it was necessary to incorporate a parameterization of subglacial discharge effects (Jenkins, 2011) on basal melting with plume theory (Jenkins, 1991; Lazeroms et al., 2018) to achieve reasonable agreement with satellite-derived basal melt rates, as two-dimensional plume theory on its own underestimated localized high melting near an inferred subglacial channel.

While several studies have implemented subglacial discharge into Antarctic regional domains and studied its effects on circulation beneath ice shelves, basal melting, and continental shelf properties (Nakayama et al., 2021; Goldberg et al., 2023; Gwyther et al., 2023), the current understanding of melt-rate sensitivities to subglacial discharge is, with exception of the work of Wekerle et al. (2024), primarily based on plume theory (Jenkins, 2011) and on idealized studies focused on a Greenland-like, vertical ice front (Xu et al., 2012, 2013; Kimura et al., 2014; Slater et al., 2015) or a narrow ice tongue (Cai et al., 2017; Wiskandt et al., 2023). Jenkins (2011) included subglacial discharge in non-rotational, one-dimensional plume theory. Using dimensional analysis he found that melt rate, within some distance from the freshwater source, has a linear dependence on temperature, and one-third power dependence on subglacial discharge. These theoretical results were to some extent reproduced in non-rotational idealized numerical simulations of melting, two-dimensional vertical faces (Xu et al., 2012; Sciascia et al., 2013), two-dimensional ice tongues (Cai et al., 2017; Wiskandt et al., 2023), and three-dimensional vertical faces (Xu et al., 2013; Kimura et al., 2014). These studies typically find a sub-linear, power-law relationship between melt rates ($\dot{m}$) and subglacial discharge ($F_s$) of the form $\dot{m} \sim F_s^n$ with $n < 1$. While Xu et al. (2012) found $n = \frac{1}{3}$, in line with theory, the remaining idealized, two-dimensional studies found slightly stronger melt-rate dependence on subglacial discharge; Sciascia et al. (2013) found $n = 0.5$, Cai et al. (2017) found $n = 0.57$, and Wiskandt et al. (2023) found $n$ between 0.41 and 0.47 for different fjord temperatures. The three-dimensional study of Xu et al. (2013) also found stronger melt-rate dependence on subglacial discharge, with $n$ between 0.5-0.9, as opposed to the theoretical one-third power. Kimura et al. (2014) found a more complex functional relationship with melt rate saturating once a critical discharge rate has been reached and the resulting high inflow velocity forced the plume away from the ice face. Wekerle et al. (2024) found $n = \frac{1}{2}$ in a three-dimensional, global realistic configuration focused on the 79 North Glacier Tongue that included rotation. However, in their study shelf conditions could evolve as a function of discharge given their configuration was global and shelf conditions were not restored, allowing for conflation of the discharge and shelf temperature effects on melting. Finally, Jackson et al. (2022) addressed the relationship between basal melting and discharge observationally. Their estimates from a relatively warm, Alaskan glacier showed that

plume theory underestimated melting by over an order of magnitude, but the one-third power dependence of melt rate on discharge seemed to hold, although with substantial uncertainties.

The question of how the horizontal distribution of subglacial discharge along grounding lines affects melt rates has also been addressed before to some extent. Kimura et al. (2014) considered the case for an unstratified water column and found that two channels in close proximity produced higher melt rates than a single channel with the equivalent total discharge. Slater et al.

(2015) considered a more realistic, stratified setup, in which plumes can reach neutral buoyancy, and found that distributed discharge always produces higher total melt rate than the equivalent channelized discharge.

Crucially, what the convectively forced plume model and the existing idealized numerical studies have neglected is the effect of the Earth's rotation, as have all other common basal melt-rate parameterizations used to force ice-sheet models (Burgard et al., 2022), including those that incorporate subglacial discharge (Pelle et al., 2023). The omission of Earth's

rotation is typically justified based on the ratio of fjord or ice-tongue width ($W$) to the first-mode baroclinic deformation radius ($R_d$) being smaller than or close to unity. However, Jackson et al. (2018), focusing on shelf-driven forcing, shows that even for $W/Rd > 0.5$, which includes most Greenland's fjords, three-dimensional dynamics are integral to understanding the fjord circulation. The omission of Earth's rotation is certainly not appropriate for Antarctic-like, ice-shelf cavities, where $W/Rd \gg 1$. In this paper, we revisit the relationship between submarine melting and subglacial discharge for a rotating,

Antarctic-like configuration. The simulations are idealized and their purpose is to provide insight into melt-rate sensitivities to subglacial discharge that, in a realistic global configuration, would be computationally expensive and potentially challenging to interpret. The idealized simulations are performed using relatively coarse resolution compared to the Greenland-like, vertical ice-cliff studies, in line with the goal of understanding sensitivities in realistic, global model configurations.

## 2   Methods

For idealized testing, we use the ocean component of the Energy Exascale Earth System Model (E3SM), the Model for Prediction Across Scales-Ocean (MPAS-Ocean; Ringler et al., 2013). MPAS-Ocean is a finite-volume, ocean model that solves the hydrostatic Boussinesq equations. It uses a horizontal mesh defined by a centroidal Voronoi tesselation. The simulations presented here use a planar mesh, rather than one on the surface of a sphere. In our experimental setup, parameterizations and coefficient choices follow the Ocean0 configuration (Figure 1) from the second Ice Shelf-Ocean Model Intercompari-

son Project, ISOMIP+ (Asay-Davis et al., 2016). We use the z* vertical coordinate (Adcroft and Campin, 2004) in the open ocean. Beneath ice shelves, the coordinate follows the ice draft, and at the seafloor, layers are dropped when they intersect the bathymetry. As a result the thickness of the top vertical layer, in contact with the ice-shelf base, varies smoothly from thinnest near the grounding line ($\sim 0.75$m), thickening towards the ice-shelf front ($\sim 15$m), and it is thickest in the open ocean ($\sim 21$m). The Ocean0 experiments presented here are performed with 36 vertical levels and 2-km horizontal resolution. Sensitivity runs

with 1-km and 4-km horizontal resolutions and 72 vertical levels were also run and produced qualitatively similar results. The change in resolution can be largely compensated for through an appropriate change in viscosity. For mixing, we employ horizontal and vertical Laplacian diffusivities and viscosities with values from Table 4 in Asay-Davis et al. (2016) that had been

chosen to be sufficiently low not to suppress eddies, but high enough to avoid numerical instabilities. There are two vertical diffusivity and viscosity values, one for stable, and one for unstable vertical profile. Horizontal diffusivity and viscosity are constant. We use the linear equation of state. The ice-shelf geometry is static. Ice-shelf basal melting is calculated using the three-equation parameterization with standard coefficients (Jenkins et al., 2010), except for the heat transfer coefficient that was tuned to give a prescribed melt rate as per the ISOMIP+ design document (Asay-Davis et al., 2016). Following Losch (2008), temperature and salinity for calculating thermal driving for ice-shelf basal melting are found by vertically-averaging over cells within 10 m of the ice base. In our vertical coordinate system, this Losch layer is necessary for melt-rate convergence with increasing vertical resolution (Gwyther et al., 2020). The Losch layer also acts as an additional source of vertical mixing near the ice base, effectively forming a mixed boundary layer of constant, spatially uniform thickness, which is used by the basal melt-rate parameterization. Heat and freshwater fluxes associated with basal melting are distributed into the ocean based on an exponential profile, with a decay length scale of 10 m, which effectively acts as enhanced vertical diffusivity near the boundary. Frazil ice is allowed to form, but it is not advected with the flow. At each time step, the model checks for frazil ice formation in the top 100 m of the water column, and any frazil mass is brought to the surface immediately and added to interfacial basal melting. The rotating cases use an $f$ plane configuration with the Coriolis parameter of $f = -1.409 \cdot 10^{-4}\,\mathrm{s}^{-1}$, which corresponds to the latitude of 75° South.

Initially, potential temperature and salinity are prescribed as horizontally uniform and linearly increasing with depth, with surface values of $T_s = -1.9$°C and $S_s = 33.8$ PSU in all cases. In the base case, the properties at the deepest point are $T_b = 1$°C and $S_b = 34.7$ PSU. At the northern boundary, potential temperature and salinity are restored to the initial values throughout the simulations. No slip boundary conditions are applied at the side walls. Subglacial discharge into the ocean at the grounding line is implemented as a volume flux of freshwater at the local pressure-dependent freezing point. Because there is no prescribed evaporation, sea level in the idealized domain increases over time due to the flux of subglacial discharge and ice-shelf basal melt into the domain. The circulation within the ice-shelf cavity is entirely driven by buoyancy forcing from ice-shelf basal melt and grounding-line freshwater flux. The simulations were run for two years, by which time the melt-rate pattern developed and stabilized. Analysis is based on the last monthly mean outputs from the two-year simulations.

In the first set of experiments we study melt-rate response to varying $F_s$, the spatially integrated subglacial freshwater flux, and how this response changes with different $F_s$ distributions along the grounding line in a rotating framework. We prescribe the boundary shelf conditions as in the base case ($T_b = 1$°C and $S_b = 34.7$ PSU). We then vary total $F_s$ between 0 and 720 m$^3$ s$^{-1}$. For each tested $F_s$ value, we also vary its horizontal release location (Figure 1e) as follows:

- (L) $F_s$ is distributed along a line corresponding to the deepest part of the grounding line

- (PW) $F_s$ is applied at a point, in the western most grid cell of L (in the case described immediately above)

- (PC) $F_s$ is applied at a point, in the center grid cell of L

- (PE) $F_s$ is applied at a point, in the eastern most grid cell of L

This set of experiments was also carried out in a non-rotating framework, setting $f = 0$, for comparison of our findings with those from previous studies.

The next set of experiments tests melt-rate sensitivity to varying amount of $F_s$ across different potential temperatures. We keep the surface temperature fixed at -1.9°C, and vary the sea floor potential temperature $T_b$ between -1.9°C and 4°C. The temperature profile was kept linear, and salinity was adjusted to keep the same density profile in all simulations. This sensitivity

test was performed for a distributed discharge (L) in a rotating framework.

Simulations were run with $T_b$ values of -1.9, -1, 0, 1, 2, and 4°C, and with $F_s$ values of 0, 7.2, 72, 180, 360, 540 and 720 $m^3\,s^{-1}$, to capture sensitivities and regimes of interest. Not every combination of $T_b$, and $F_s$ was run for every horizontal release location. $T_b$ values were chosen to include present day and plausible future range of temperatures occupying the continental shelves of Antarctica. Similarly, $F_s$ values were chosen to span the relatively wide range of modeled subglacial discharge

regimes around Antarctica (e.g., Hager et al., 2022b).

By design, some models, such as MPAS-Ocean, are required to have more than a single layer at the grounding line. This then brings up the modeling choice of where to apply $F_s$ in the vertical. In the above experiments we chose to distribute $F_s$ uniformly in the vertical. Melt-rate sensitivity to this modeling choice is addressed in Appendix A.

## 3 Results

With $F_s = 0$, the total basal melt flux over the ice shelf increases quadratically with the far-field potential temperature (Figure A1a), in agreement with previous simulations (Holland et al., 2008) and theory (Jenkins et al., 2018). By construction, melt rates are proportional to friction velocity and thermal driving. In the rotating case, the melt-rate spatial pattern is such that melting is highest at the deepest parts of the ice shelf and enhanced at the western grounding line (Figure 2a1). This holds as long as the thermal driving remains sufficient (Figure 2a3) to support melting, while the ascending meltwater plume veers

westward under the influence of Coriolis force (Figure 2a2). Friction velocity shown here in figures (e.g., Figure 2, second column) is a good proxy for plume speed as long as the plume is attached to the ice base.

The interplay between friction velocity and thermal driving, when $F_s$ is introduced, is shown in Figure 2 for the rotating case and in Figure 3 for the non-rotating one. The sections shown are for the moderate discharge strength of $F_s = 72\ m^3/s$, however, the results do not change qualitatively with the strength of $F_s$ within the tested range of values. $F_s$ enhances friction velocities

(e.g., Figure 2e2 and Figure 3e2) due to added buoyancy, increasing melt that cools the boundary layer (e.g., Figure 2e3 and Figure 3e3), which in turn suppresses melt. Near the $F_s$ source, the friction velocity increase is so high that it dominates any cooling, producing positive melt-rate anomalies; however with increasing distance from the source, melt-driven cooling starts compensating the heightened friction velocity, resulting in little change in melt rates (e.g., Figure 2e1 and Figure 3e1). The plume slows down quickly as it rises, entrains ambient waters, and thickens. In the rotating case, the plume veers westward

relative to its direct, upslope and northward direction with no rotation. The result is meltwater accumulation near the western grounding line (Figure 4a2), compared to the east (Figure 4a1), and increased overall stratification with lighter waters at the surface and meltwater extending deeper into the water column in the west (Figure 4a2). This stratification asymmetry, present

already in the reference case with rotation, sets the stage for the observed qualitative differences in melt response when $F_s$ is introduced and its release location is varied along the grounding line (Figure 2, left column).

To quantify the rotational effects on the bulk scaling relationship between basal melting and $F_s$, we introduce $\Delta\dot{M}$, the anomaly in spatially integrated basal melt volume flux with respect to the reference case with $F_s = 0$. We use total melt-flux anomalies rather than spatially averaged melt-rate anomalies because for relatively localized anomalies, as is the case here, the latter would introduce a more evident dependence of the results on ice-shelf area. As the analysis is focused on the last monthly mean output (of 24 months of simulation) $\Delta\dot{M}$ is also averaged in time. We observe that, in the presence of rotation, $\Delta\dot{M}$ is strongly dependent on the horizontal release location of point source $F_s$ (Figure 5a), especially compared to the non-rotating case (Figure 5b).

## 3.1 Melt sensitivity to horizontal discharge release location

Starting with the rotating case, the point-source experiments (PW, PC, and PE) show that the most effective location to apply $F_s$ in terms of $\Delta\dot{M}$ is at the eastern portion of the grounding line; $\Delta\dot{M}$ more than doubles when $F_s$ is applied in the east, rather than in the west (Figure 5a). When an equivalent $F_s$ is distributed along the grounding line (L), the resulting $\Delta\dot{M}$ is approximately halfway between the $\Delta\dot{M}$ in the PW and PE cases, and is slightly higher than in the PC case. $\Delta\dot{M}$ in the PE case is about 1.5 times higher than in the distributed (L) case. This is in part due to a larger portion of the ice-shelf area sustaining positive melt-rate anomalies when the plume is located farther east on the grounding line (Figure 2e1) compared to its central (Figure 2d1) or western location (Figure 2c1). Consistent with the point-source experiment, the distributed case experiences a larger spatial extent of melt-rate increase in the east than in the west (Figure 2b1).

When $F_s$ is released at a point in the east (PE), where the water column beneath the ice shelf is less stratified compared to the west (Figure 4c1 vs 4c2), the added freshwater induces high flow speeds, cooling, and freshening in a thin layer near the ice base, with relative warming beneath it (Figure 6e1). This fast thin outflow near the ice base persists as the plume rises, spreads laterally, and turns west (Figure 6e2). However, along the eastern and central transects no portion of the plume has reached neutral buoyancy as it does in the non-rotating case (Figure A2e1). Instead, the plume becomes neutrally buoyant in the western part of the domain, where the presence of the western boundary forces the plume to flow upslope, rise and mix, as visible in the large temperature anomaly throughout the western section (Figure 6e3). The addition of $F_s$ has increased vertical mixing in the west, as evidenced by relative warming at the top and relative cooling at the bottom of the water column.

When $F_s$ is released at a point in the center of the grounding line (PC), the response downstream of the release location is similar to the PE case, only weaker (Figure 6d vs Figure 6e). The friction velocity increase is lower in this case (Figure 2d2 vs Figure 2e2) because the water column is more stratified here, having already been freshened by meltwater coming from the east. This also means, that the density difference between the subglacial discharge and the surface waters is reduced, as is the driving force for the plume ascent. Finally, the presence of the western boundary means that, the closer the discharge location is to the west, the less distance the plume is able to travel across the ice shelf before it encounters the boundary and is forced upward along the ice base (Figure 2, second column).

When $F_s$ is released at a point near the western boundary (PW), these tendencies are even more amplified. The water column is more stratified here, the surface water along the ice base fresher and lighter (Figure 4b2), and the proximity of the western boundary closer, which forces upward flow of meltwater almost immediately upon release (Figure 2c2). The result is that the plume has less distance to travel along the ice base and it becomes neutrally buoyant deeper in the water column. This is seen as relative warming at the top and relative cooling at the bottom of the water column, much closer to the grounding line (Figure 6c3). Unlike in the PE and PC cases, for PW this relative warming at the top extends as far as the boundary layer, causing a warm anomaly in the thermal driving (Figure 2c3).

In summary, the east-west asymmetry in meltwater distribution means that $F_s$ added in the west inputs less buoyancy, and therefore less available potential energy into the system than $F_s$ added in the east. Together with the east-west asymmetry in stratification, this means that when injected in the west the plume becomes neutrally buoyant deeper in the water column, keeping any positive melt-rate anomalies closer to the grounding line and lowering their magnitude (Figure 2, left column).

## 3.2 Scaling relationships between subglacial discharge and melt

We now turn our attention to the scaling relationship between melt flux and $F_s$, which has been extensively studied for non-rotating scenarios. Figure 5 not only shows that $\Delta \dot{M}$ depends on horizontal distribution of $F_s$ for the rotating case with point source discharge, but it also indicates that the functional form of the sub-linear relationship between $\Delta \dot{M}$ and $F_s$ differs depending on whether rotation is included and whether $F_s$ is distributed or channelized (Figure 5).

To investigate how the functional form of the sub-linear relationship between $F_s$ and $\Delta \dot{M}$ depends on rotation for the channelized drainage system, we first remove differences associated with the domain asymmetries and the presence of boundaries. To do that, we average the $\Delta \dot{M}$ results for all three point release cases separately for the rotating and the non-rotating scenarios (Figure 5). We confirm that the non-rotating case complies with theory, that is, $\Delta \dot{M}$ scales with one-third power of $F_s$ (Figure 7a, dashed black line), as in Jenkins (2011). However, in the rotating case, the one-third power scaling clearly overestimates $\Delta \dot{M}$ for low $F_s$ and underestimates it for high $F_s$ (Figure 7a, dashed red line), which means that the curvature of the one-third power scaling is incorrect. A higher power scaling seems to be a better fit for the rotating case; we find that $\Delta \dot{M}$ scales well with two-thirds power of $F_s$. We note that the stronger $\Delta \dot{M}$ scaling with $F_s$ for the rotating case does not depend on $f$ as long as $W/Rd \gg 1$ (Figure A1c).

Nevertheless, the theory of Jenkins (2011) should still hold in the rotating case at small distances from the source before the Coriolis effect becomes important. To test this, instead of integrating over the entire domain, we integrate melt-flux anomalies over a fixed distance from the $F_s$ source. We compare $\Delta \dot{M}$ integrated over the whole domain, over an area within 10 km from the source, and within 2 km from the source (Figure 7a-c). We observe that, for the rotating case, the scaling between $\Delta \dot{M}$ and $F_s$ becomes closer to one-third power as melt fluxes are integrated over a smaller area nearer to the source (Figure 7c), as expected from the non-rotating, convectively forced plume theory.

To further investigate possible causes of the power law difference due to the inclusion of rotation, we partition the ice-shelf area into a high-melting, plume-dominated region and a low-melting, ambient-dominated region, and study how each partition contributes to the overall melt-flux anomaly. To capture the high-melting region near the $F_s$ source and along the path of the

elevated plume-driven friction velocities in each case, we define the plume-dominated partition as the region where the melt-rate anomaly exceeds the threshold of 10% of the maximum melt-rate anomaly, and the ambient-dominated partition as the remainder of the ice shelf (Fig. 8). (We considered various criteria for the partitioning, and found the 10% threshold to be most representative for channelized drainage.) Because most of the melt-rate anomaly occurs within a few tens of kilometers of the grounding line, we restrict our attention to the area within 30 km from the $F_s$ source (shown in Figure 8 for the PC case).

As done previously, for both the rotating and the non-rotating scenarios, we average the results from all three point release cases (PE, PC, and PW). We quantify the relative contribution of the two partitions to the total melt-flux anomaly, and inspect the change of the partition area and of mean melt-rate anomaly, averaged over each partition separately, as $F_s$ increases. The melt-flux anomaly $\Delta\dot{M}$ (e.g., Figure 9a) is the product of mean melt-rate anomaly (e.g., Figure 9b) and the area over which the melt-rate anomaly is averaged (e.g., Figure 9c). The partitioning changes from case to case (Fig. 8), therefore the melt-rate

anomaly is averaged, and the melt-flux anomaly integrated, over a different area for each partition in each scenario.

We observe that most of the melt-flux increase with $F_s$ occurs over the plume-dominated partition in both rotating and non-rotating cases (Figure 9a, d). Therefore, explaining the melt-flux anomaly behavior of the plume-dominated partition is sufficient to the understanding of the total melt-flux anomaly behavior as $F_s$ increases. We find that while the melt-flux anomaly respects the power law scaling identified above, $n = \frac{2}{3}$ with rotation and $n = \frac{1}{3}$ without (Figure 9a, d), the melt-rate anomaly

averaged over the plume-dominated partition scales with one-third power of $F_s$ both with and without rotation (Figure 9b, e). This is reconciled by the observation that the area fraction of the plume-dominated partition scales with one-third power of $F_s$ in the presence of rotation (Figure 9c), but is nearly constant in the non-rotating case (Figure 9f). This means that it is the increase in the area of the plume-dominated partition that causes stronger dependence of the melt-flux anomaly on $F_s$ when rotation is included. By design, the plume-dominated partition (Figure 8a, right column) closely follows the plume path

identified by high friction velocities (Figure 8a, left column). Therefore, it is clear that the higher area increase of the plume-dominated partition with $F_s$ is related to the increasing distance over which the plume is deflected by Coriolis force as its flow speed increases. While for the rotating case the plume path increases with the strength of $F_s$, this is not the case for the non-rotating case, where the flow is upward along the ice base independent of $F_s$. There, as the plume cools with increased melt near the discharge location, an initially sharp melt-rate increase quickly declines with distance (Figure 8b, middle column),

resulting in little area change of the plume-dominated partition (Figure 8b, right column).

Turning to the distributed system and repeating the same analysis, we observe that $\Delta\dot{M}$ scales with two-thirds power of $F_s$ for both the rotating and the non-rotating case (Figure A3a, solid lines). When averaged over cells within 2-km of the source, that is one grid-cell distance away from the $F_s$ source, the $\Delta\dot{M}$ scaling with $F_s$ lies between one-third and two-thirds power (Figure A3c), but already when averaged over 10-km distance from the $F_s$ source, the scaling is much closer to two-

thirds power for both cases (Figure A3b). The analysis that partitions the ice-shelf area within 30 km from the $F_s$ source into plume-dominated and ambient-dominated regions shows similar results for the rotating and non-rotating scenarios. Most of the melt-flux anomaly with $F_s$ occurs over the plume-dominated partition in both rotating and non-rotating cases (Figure A4a, d). In both cases the melt-rate anomaly averaged over the plume-dominated partition (Figure A4b, e) and the area fraction of the plume-dominated partition (Figure A4c, f) scale with $F_s$ similarly, such that their product gives a two-third power dependence

of the melt-flux anomaly on $F_s$ (Figure A4a, d). For the distributed system there is some sensitivity of the exact scaling relation of both the area fraction and the associated melt-rate anomaly with $F_s$ to the choice of the partitioning criteria, but the result – that these scalings are the same with and without rotation – is independent of the threshold.

## 4 Discussion

From our idealized experiments that include rotation, we found that channelized discharge can either increase or decrease the integrated melt-flux anomaly compared to the distributed system, in our particular setup by about 50% (Figure 5a). This contrasts with findings from non-rotating experiments by Slater et al. (2015). They modeled a Greenland-like vertical ice front and varied the distribution of freshwater discharge along the grounding line, concluding that distributed $F_s$ is always more efficient in driving melting than channelized $F_s$, in their simulations by up to a factor of five. Results from our non-rotating experiments are qualitatively consistent with Slater et al. (2015), and we observe a higher $\Delta \dot{M}$ for horizontally distributed than for channelized discharge. This is particularly pronounced as $F_s$ increases, owing to the higher exponent in the power-law relationship between $\Delta \dot{M}$ and $F_s$ in the distributed case than in the channelized case (Figure 5b). A key reason the findings of Slater et al. (2015) – that melt-rate response to discharge increases with the number of sources – do not apply in the rotating case is that they rely on individual plumes not interacting with each other. In the non-rotating case, individual plumes only interact when spaced sufficiently close together, but in the rotating case the plumes interact even if spaced far from each other. Rotation eventually makes all melt from individual plumes coalesce on one side of the domain, which results in less ice-shelf area available for discharge-driven melt-rate increase compared to the non-rotating case.

The reason for the two-thirds power dependence of melt flux on $F_s$ in the non-rotating, distributed case compared to the one-third power dependence in the channelized, non-rotating case is not clear, but appears to be related to the faster growth of the area of the plume-dominated partition with $F_s$ in the distributed case and may be related to interacting plumes. Although this difference has not been explicitly discussed in previous studies, it may be implicitly present in the work of Xu et al. (2013). They studied the relationship between submarine melt rate and discharge through three-dimensional simulations of a vertical ice front, finding a more complicated, sub-linear relationship than a simple power law, and identified two regimes, one appropriate for low $F_s$ and the other for high $F_s$. In the light of our results, this difference can be attributed to the way variations in $F_s$ magnitude were prescribed in their study; Xu et al. (2013) fixed the value of $F_s$ in each grid cell and, to increase total discharge, they increased the number of grid cells where this fixed $F_s$ was applied. Thus, for lower values of $F_s$, they effectively simulated a channelized system and, for higher values of $F_s$, the system effectively became distributed. As such, according to our results, the power law gradually changes from a one-third to a two-thirds power dependence, giving rise to the more complicated functional relationship reported on in Xu et al. (2013). Notably, the parameterization of Xu et al. (2013), subsequently adopted by the Ice Sheet Model Intercomparison Project for CMIP6 - Greenland Ice Sheet (ISMIP6-GIS; Slater et al., 2020), is widely used in ice-sheet models as the standard vertical ice-cliff melt-rate parameterization. While we are not able to simulate vertical cliffs, our results from non-rotating ice-shelf simulations suggest the functional form of the melt-rate dependence on discharge from Xu et al. (2013) to be revised. Specifically, rather than the exponent ($n$) being a function of

the discharge strength, our simulations indicate that $n$ depends on whether the discharge is distributed ($n = \frac{2}{3}$), or channelized ($n = \frac{1}{3}$).

Our results have implications for melt-rate parameterizations that attempt to realistically account for the effects of subglacial discharge on ice-shelf basal melting. The same amount of total discharge can more than double $\Delta \dot{M}$ if located in the east as opposed to the west of the domain (for the Southern Hemisphere). This result is based on a highly idealized ice-shelf domain, and will likely change quantitatively as additional geometric complexities are included. The sensitivity of $\Delta \dot{M}$ to discharge location along the grounding line is not accounted for in existing parameterizations (Jenkins, 2011; Pelle et al., 2023), and

may be difficult to include accurately, especially for more complicated geometries. Our results highlight the sensitivity of melt rates to the location of channelized, submarine discharge and, as such, also highlight the importance of accurate modeling of subglacial meltwater discharge via subglacial hydrology models.

     We have primarily focused on characterizing spatially integrated melt-flux anomalies in the presence of subglacial discharge, as have previous ocean modeling studies of melting vertical cliffs and floating ice tongues. While some ice-sheet modeling

studies indicate that spatially averaged melt rates are far more important for determining ice-sheet dynamics than their spatial variations (Joughin et al., 2024), a more extensive body of literature argues that melt rates near grounding lines and pinning points matter to the ice-sheet dynamics much more than melt rates elsewhere on the ice shelf (Gagliardini et al., 2010; Fürst et al., 2016; Reese et al., 2018; Zhang et al., 2020). An understanding of how the melt-rate anomalies due to subglacial discharge ultimately affect ice-sheet dynamics will require an ice-sheet model with a suitable melt-rate parameterization (e.g.,

Pelle et al., 2023) or a coupled ice-sheet/ocean/subglacial hydrology model.

## 5    Conclusions

We studied the effect of subglacial discharge on submarine melt rates for an idealized, Antarctic-like ice shelf within a rotating reference frame. This work complements existing studies focusing on non-rotating, idealized, Greenland-like vertical calving fronts and narrow ice tongues. The motivation for the study was to develop a conceptual understanding of the sensitivities and

response of the system in an idealized setup before conducting global, realistic simulations. Relevant to the realistic simulations, we found that with restoring shelf conditions the total melt-flux anomaly scales with two-thirds power of the discharge. This is a stronger melt-flux anomaly dependency on discharge than found in previously studied, non-rotating cases. It results from a larger fraction of the ice-shelf base being exposed to high melt rates, locally driven by the rising plume that is deflected under rotation and whose travel path increases when forced by stronger discharge. We also found that the melt-rate response

is strongly sensitive to the location of the discharge along the grounding line; the efficiency of subglacial discharge, in terms of total melt-flux anomaly, grows with distance from the area where meltwater ultimately accumulates due to rotational effects (the western ice-shelf margin in this study). The relationship between melt-flux anomaly and the number of sources for a given total discharge derived from non-rotating, vertical cliff experiments does not translate to the rotating ice-shelf scenario. Our results complicate the task of melt-rate parameterizations aspiring to account for subglacial discharge effects, in that there is

a significant, additional melt-flux sensitivity to discharge location. The sensitivity to the location of discharge also lends new importance to accurate modeling of subglacial hydrology.

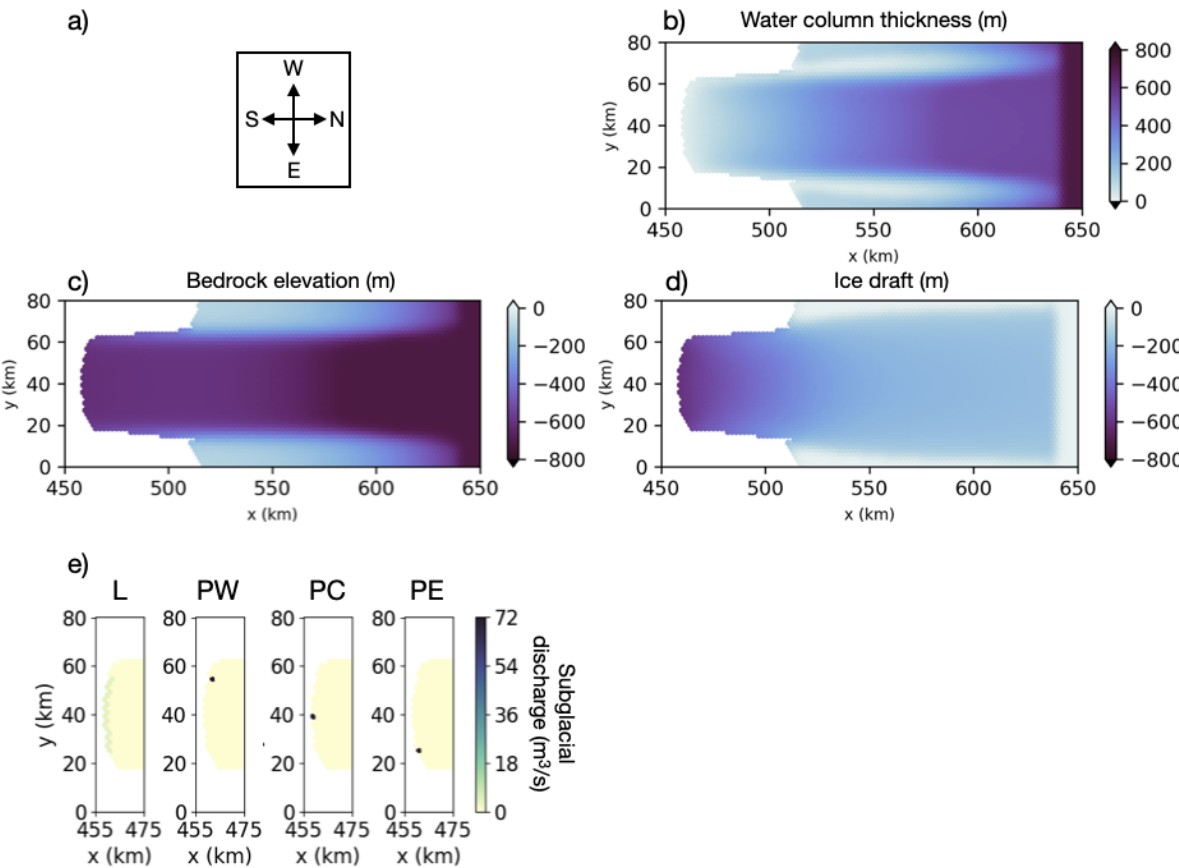

**Figure 1.** a) Convention for geographic references in text. b-d) Idealized ISOMIP+ domain setup. b) Water column thickness. c) Bedrock elevation. d) Ice draft. e) Distribution of subglacial discharge of $F_s = 72$ m$^3$/s at the grounding line along a line (L) or at a point west (PW), center (PC), or east (PE).

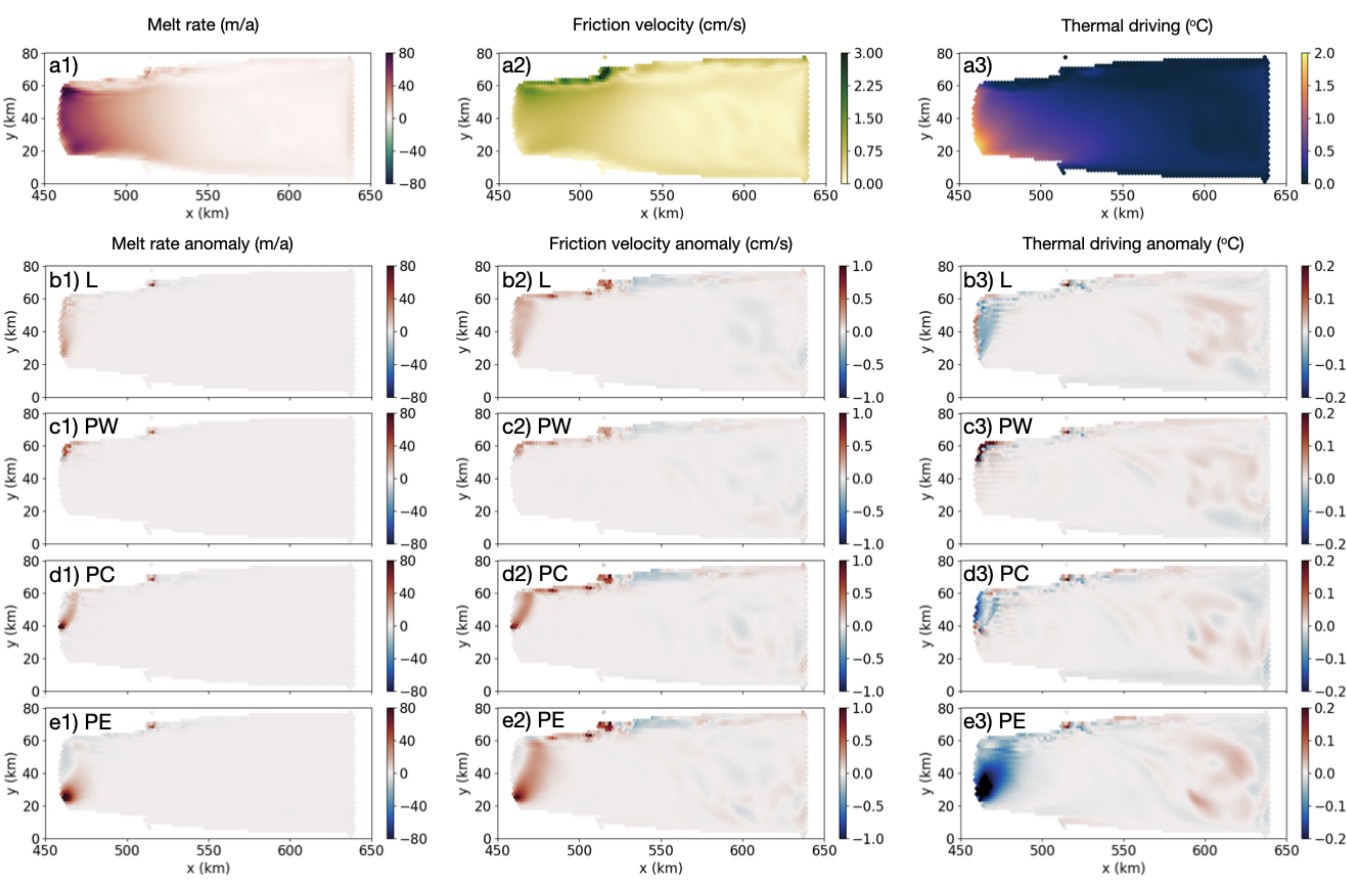

**Figure 2.** a) Melt rate, friction velocity, and thermal driving for the reference case without subglacial discharge ($F_s = 0$ m³/s). b-e) Melt-rate, friction-velocity, and thermal-driving anomalies when $F_s = 72$ m³/s is added at various locations along the grounding line. b) $F_s$ is distributed along a line (L). c-e) $F_s$) is channelized, released at c) point west (PW), d) point center (PC), and e) point east (PE).

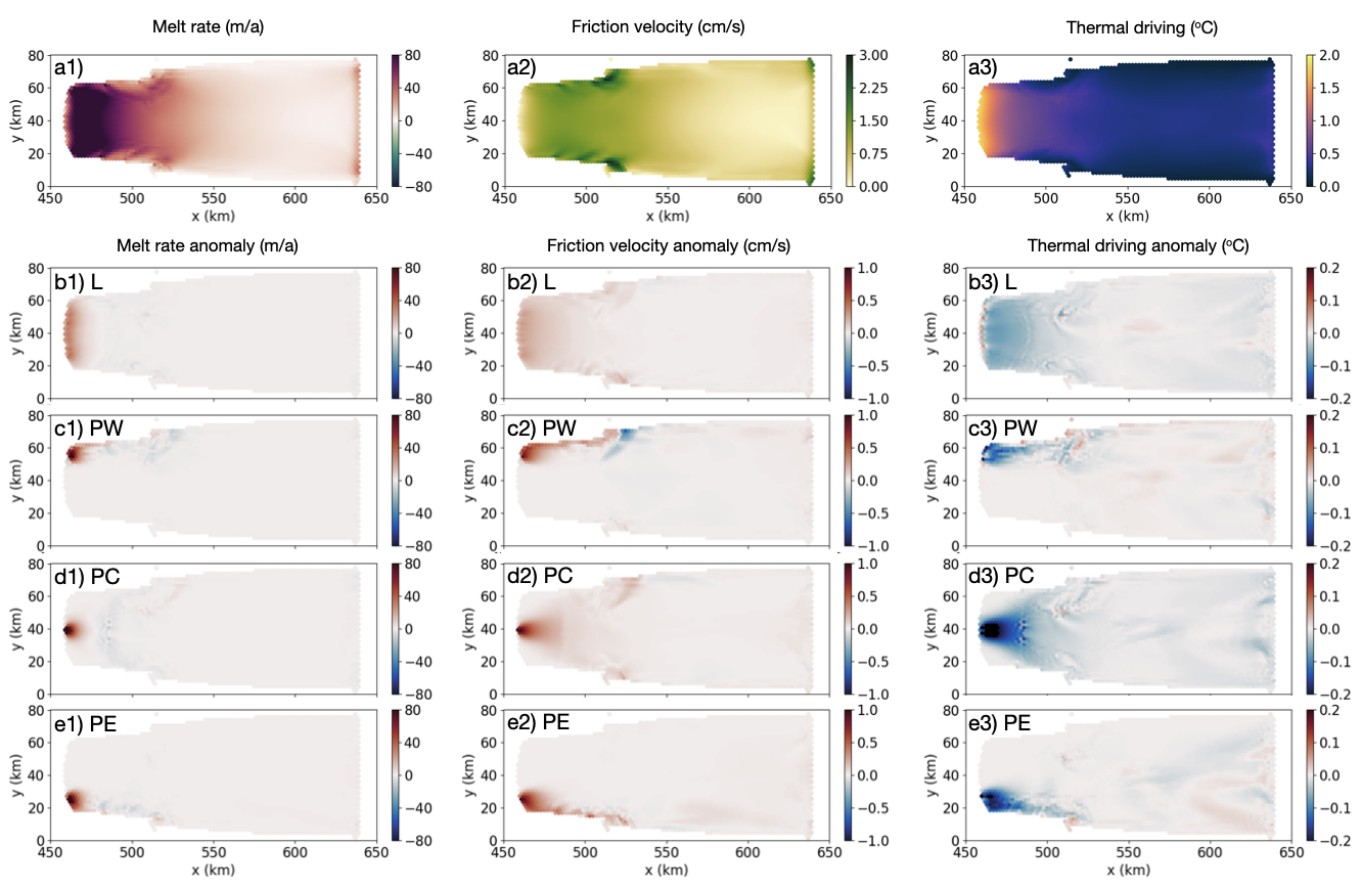

**Figure 3.** Same as Figure 2 but for the non-rotating case.

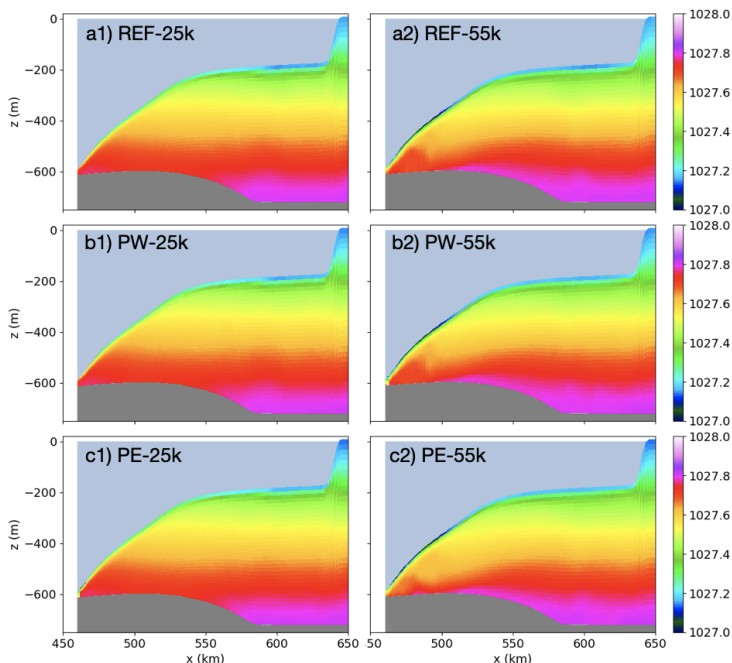

**Figure 4.** Potential density sections. a) Reference case without subglacial discharge ($F_s = 0$ m$^3$/s). b) Channelized discharge ($F_s = 72$ m$^3$/s) applied at point west (PW). c) same as b) but for point east (PE). The sections are located at y = 25 and 55 km, from left to right, corresponding to progression from east to west (Figure 1).

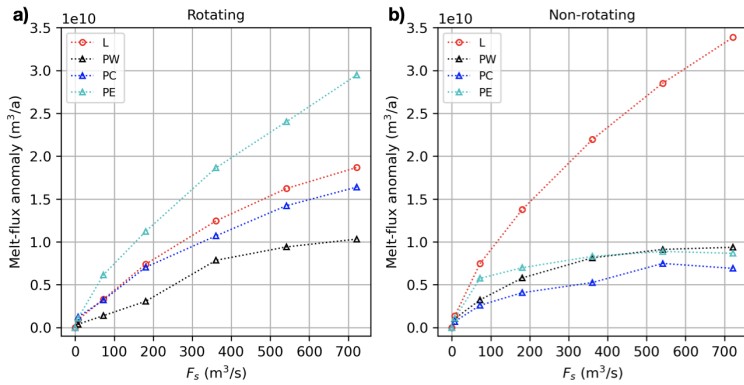

**Figure 5.** Total melt-flux anomaly ($\Delta\dot{M}$) as a function of subglacial discharge ($F_s$) and its horizontal release location and distribution. a) Rotating case, horizontal distribution sensitivity: $F_s$ is distributed along a line (L) or released at different channelized points (west - PW, center - PC, east - PE). b) Same as a) for the non-rotating case.

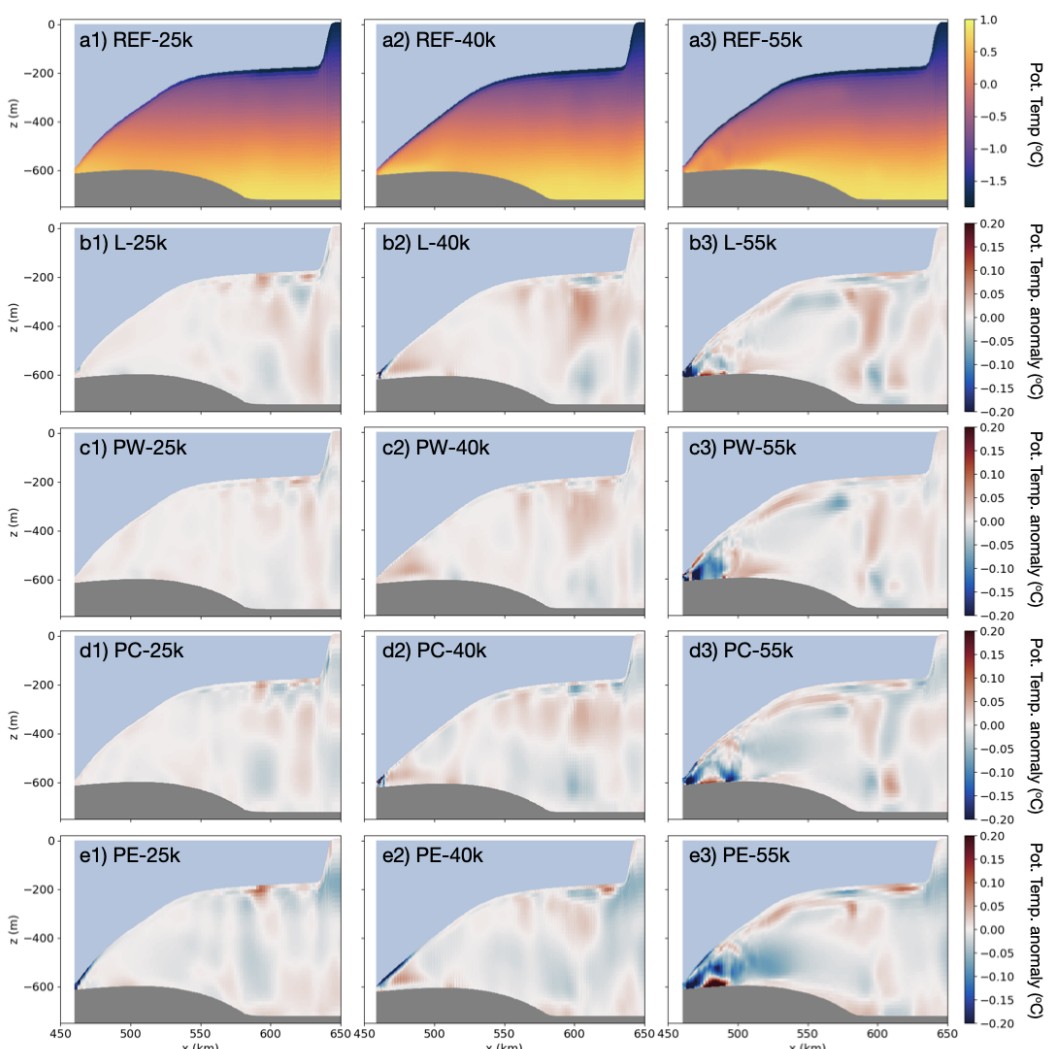

**Figure 6.** a) Potential temperature sections for the reference case without subglacial discharge ($F_s = 0$ m³/s). b-e) Potential temperatures anomalies when $F_s = 72$ m³/s is added at various locations along the grounding line. b) $F_s$ is distributed along a line (L). c-e) $F_s$) is channelized, released at c) point west (PW), d) point center (PC), and e) point east (PE). The sections are located at y = 25, 40, and 55 km, from left to right, corresponding to progression from east to west (Figure 1).

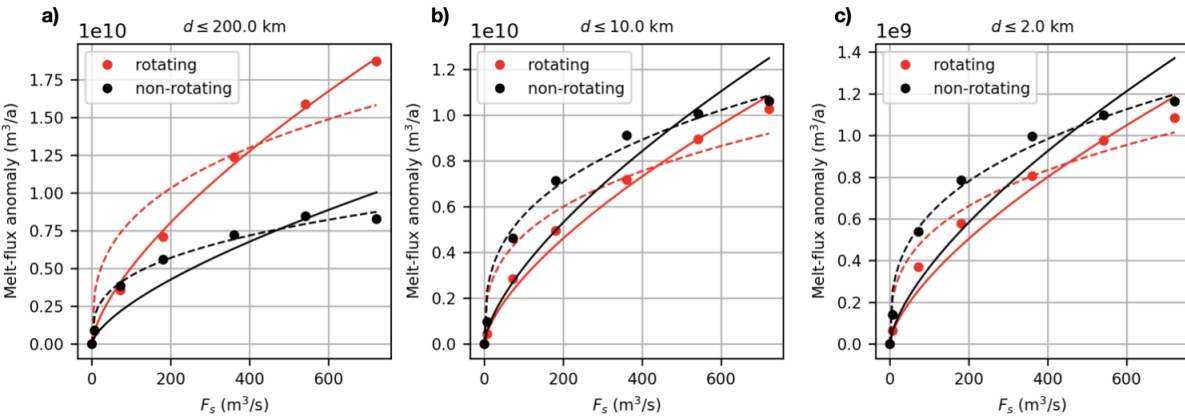

**Figure 7.** Scaling relationships between integrated melt-flux anomaly ($\Delta\dot{M}$) and subglacial discharge ($F_s$) for channelized system from averaged PW, PC, and PE experiments as in Figure 5. The melt-flux anomaly is integrated over distance $d$ from the discharge location. The dashed line is scaling with $F_s$ to one-third power and the solid line scaling with $F_s$ to two-thirds power.

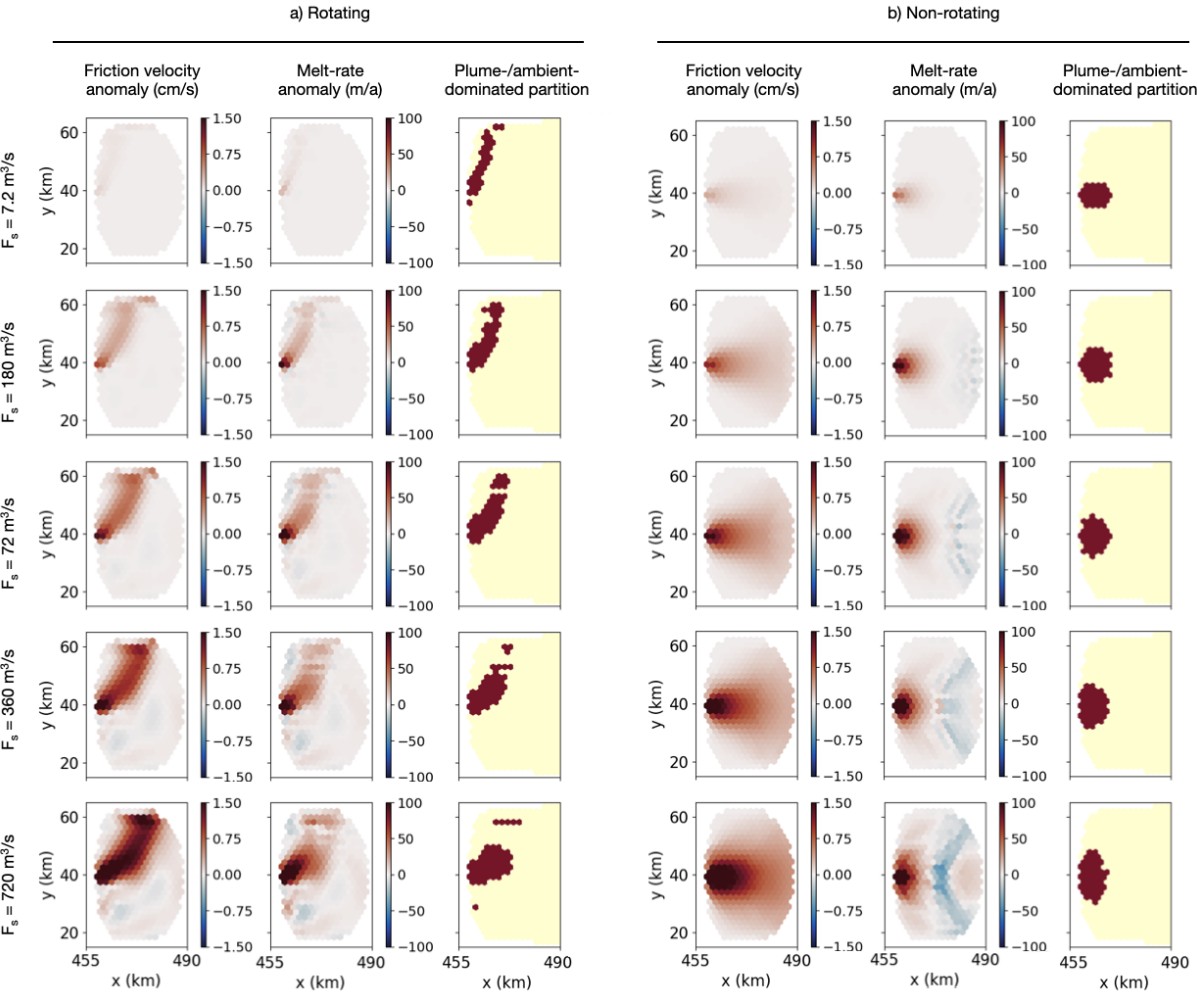

**Figure 8.** Friction velocity anomaly, melt-rate anomaly, and area partitioning into high-melting, plume-dominated (dark area) and low-melting, ambient-dominated regions (light area) for a) rotating and b) non-rotating case. We show ice-shelf area within 30 km from the $F_s$ source, which is used in the analysis in Figure 9. The rows are different values of the channelized discharge $F_s$ released at point center (PC).

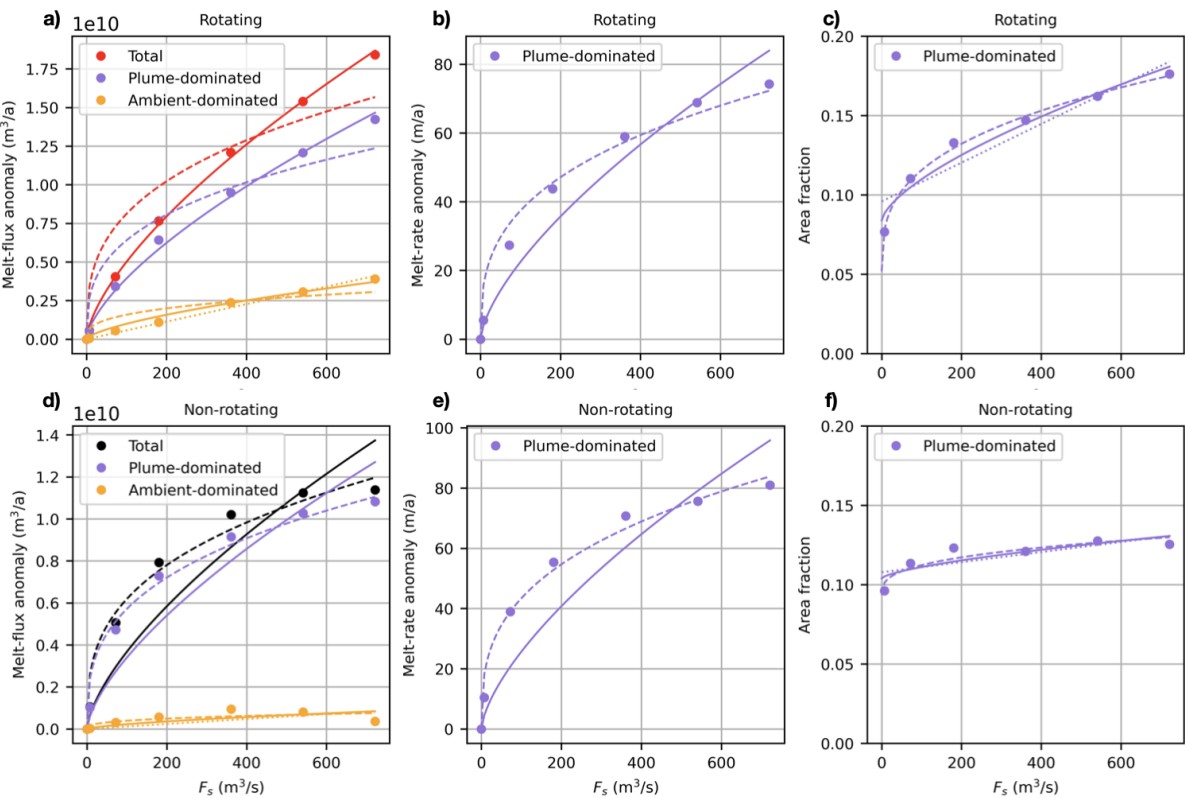

**Figure 9.** Scaling relationships with subglacial discharge ($F_s$) for different quantities evaluated over a plume-dominated or ambient-dominated partition (Figure 8) of the ice-shelf base within 30 km of the discharge location. We consider the averaged channelized experiments. a, d) Spatially integrated melt-flux anomaly. b, e) Spatially averaged melt-rate anomaly. c, f) Area fraction of an ice-shelf partition. a-c show the rotating case and d-f the non-rotating case results. In all panels the dashed line is scaling with $F_s$ to one-third power, the solid line scaling with $F_s$ to two-thirds power, and the dotted line a linear scaling with $F_s$.

*Code availability.* The E3SM code is available at https://github.com/E3SM-Project/E3SM, and the branch used for the simulations presented here is https://github.com/irenavankova/E3SM/tree/sg_pull_w_fraz (Git hash: 52b7bf50d52f7d9d4214ae656f225fb45223a2ee). The test cases were constructed and run using COMPASS (Configuration Of Model for Prediction Across Scales Setups), available at https://github.com/MPAS-Dev/compass.

## Appendix A

We tested the total melt-flux anomaly sensitivity to the modeling choice of $F_s$ vertical distribution as follows:

- (T) $F_s$ is concentrated in the top vertical layer

- (U) $F_s$ is uniformly mixed in the vertical

- (B) $F_s$ is concentrated in the bottom vertical layer

This sensitivity test was performed for a distributed discharge in a rotating framework for three different bottom potential temperatures, -1°C , 0°C, and 1°C. As expected, based on the different amounts of added available potential energy, highest $\Delta \dot{M}$ is produced when $F_s$ is released at the bottom vertical layer, and lowest $\Delta \dot{M}$ when $F_s$ is released at the top layer (Figure A1b). While the difference in $\Delta \dot{M}$ caused by a choice of vertical release location can be up to 20%, this choice does not affect the scaling relationship between $\Delta \dot{M}$ and $F_s$.

We also tested the robustness of the scaling relationship between $\Delta \dot{M}$ and $F_s$ to varying latitude via the Coriolis parameter. This was tested for the PC case for latitudes of $45°$, $65°$, $85°$, and $90°$ South, in addition to the latitudes of $0°$ and $75°$ South that are the focus of the paper. We find that in cases where $W/Rd \gg 1$, $\Delta \dot{M}$ scales well with two-thirds power of $F_s$ independently of latitude (Figure A1c).

Finally, to check whether the scaling of $\Delta \dot{M}$ with $F_s$ holds across different ambient temperatures, we run temperature sensitivity experiments for the rotating, distributed case. We find that the two-thirds power scaling holds across different values of far-field bottom potential temperature (Figure A5a), especially for interfacial melting. At colder temperatures ($T_b = -1.9°$C), when significant amount of frazil forms and frazil freezing anomalies largely compensate interfacial melting anomalies, the $\Delta \dot{M}$ sensitivity to $F_s$ drops and saturates (Figure A5a, black dashed line). Unfortunately, the representation of frazil dynamics in MPAS-Ocean is rather simplistic, so we are not in a position to investigate frazil-related dynamics in greater detail. Finally, based on the rotating, distributed case, we confirm that $\Delta \dot{M}$ depends approximately linearly on ambient potential temperature (Figure A5b) in the same way as it does in the non-rotating, convectively-forced plume theory (Jenkins, 2011). The functional fit used to produce the solid curves in Figure A5 is

$$\Delta \dot{M} = k(T_b - T_0)F_s^n, \tag{A1}$$

with $n = \frac{2}{3}$ being imposed. The remaining coefficients $T_0$ and $k$ can be obtained using regression, but their values will likely depend on the ice-shelf dimensions and shape. The least-squares fit in Figure A5 yields $T_0$ a bit lower than the freezing point,

indicating that the equation does not apply well when ambient temperatures are close to the freezing point and when frazil dynamics becomes important (Figure A5a, black dashed line), as already discussed.

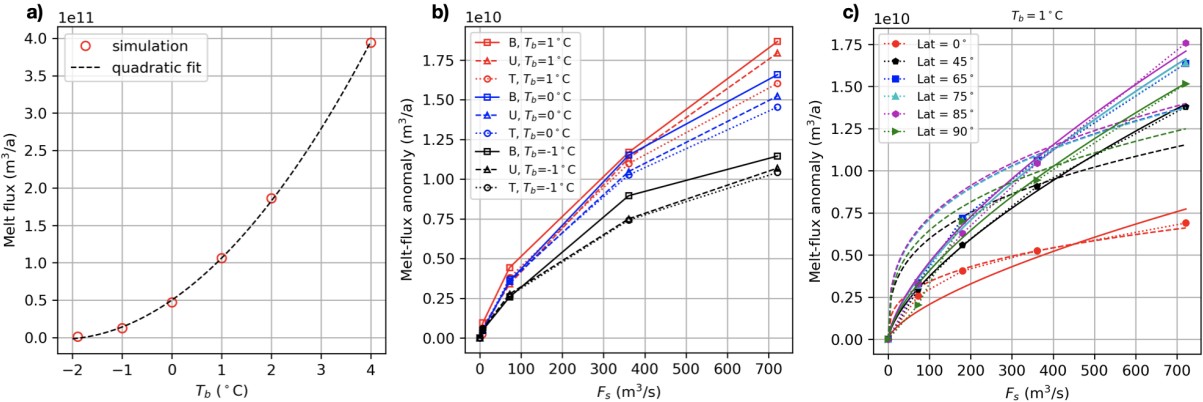

**Figure A1.** a) Mean reference basal melt flux for different shelf conditions set by property restoring at the northern boundary; $T_b$ is the sea-floor restoring potential temperature. The dashed line shows a quadratic fit. b) Total melt-flux anomaly ($\Delta\dot{M}$) as a function of distributed subglacial discharge strength ($F_s$) for different vertical release locations and temperatures. The sensitivity is tested for a rotating case, and $F_s$ is distributed along a line (L) and released at the bottom (B) layer, top (T) layer, or uniformly mixed in the vertical (U). c) $\Delta\dot{M}$ as a function of $F_s$ for varying latitude via the Coriolis parameter. The dashed line is $\Delta\dot{M}$ scaling with $F_s$ to one-third power and the solid line is $\Delta\dot{M}$ scaling with $F_s$ to two-thirds power, using least-squares fit. The filled dots connected by dotted lines are simulation results.

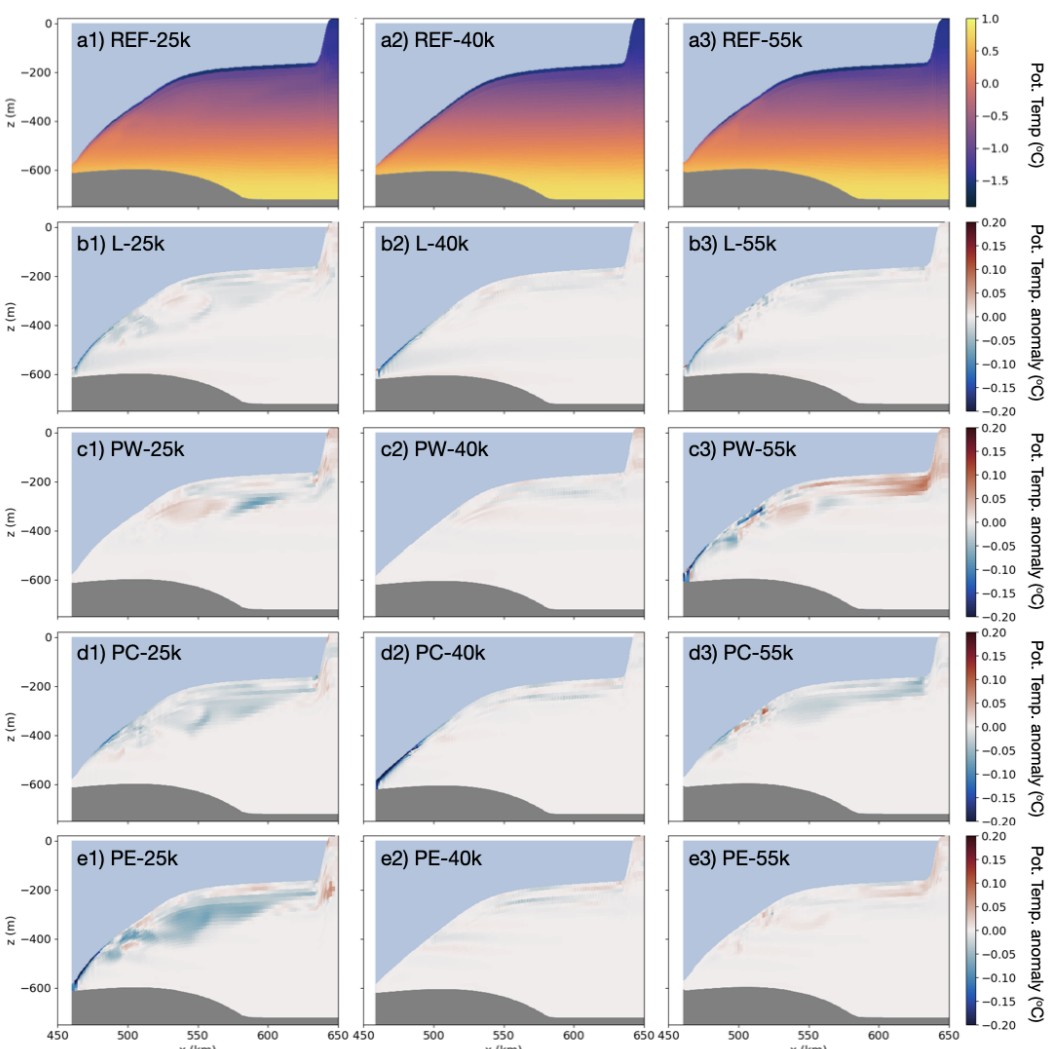

**Figure A2.** Same as Figure 6 but for the non-rotating case.

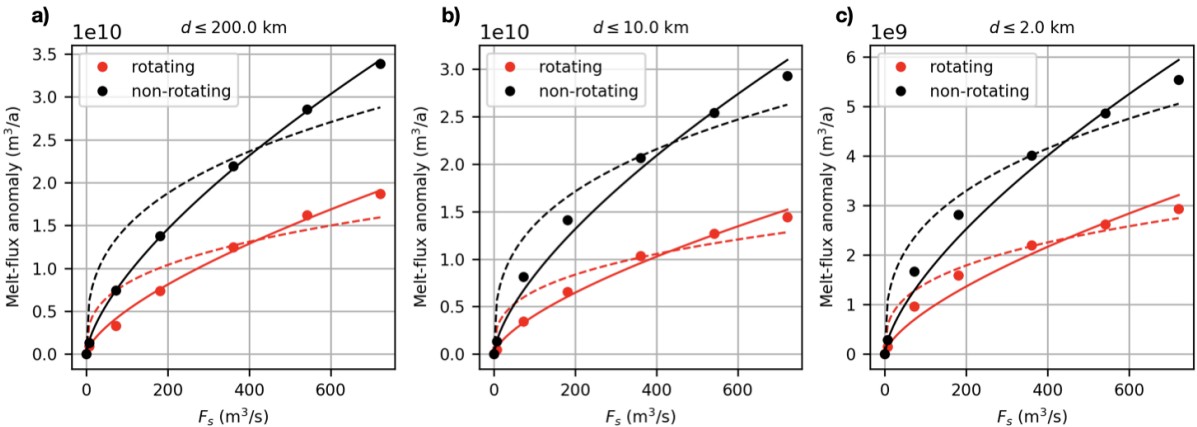

**Figure A3.** Same as Figure 7 but for distributed discharge.

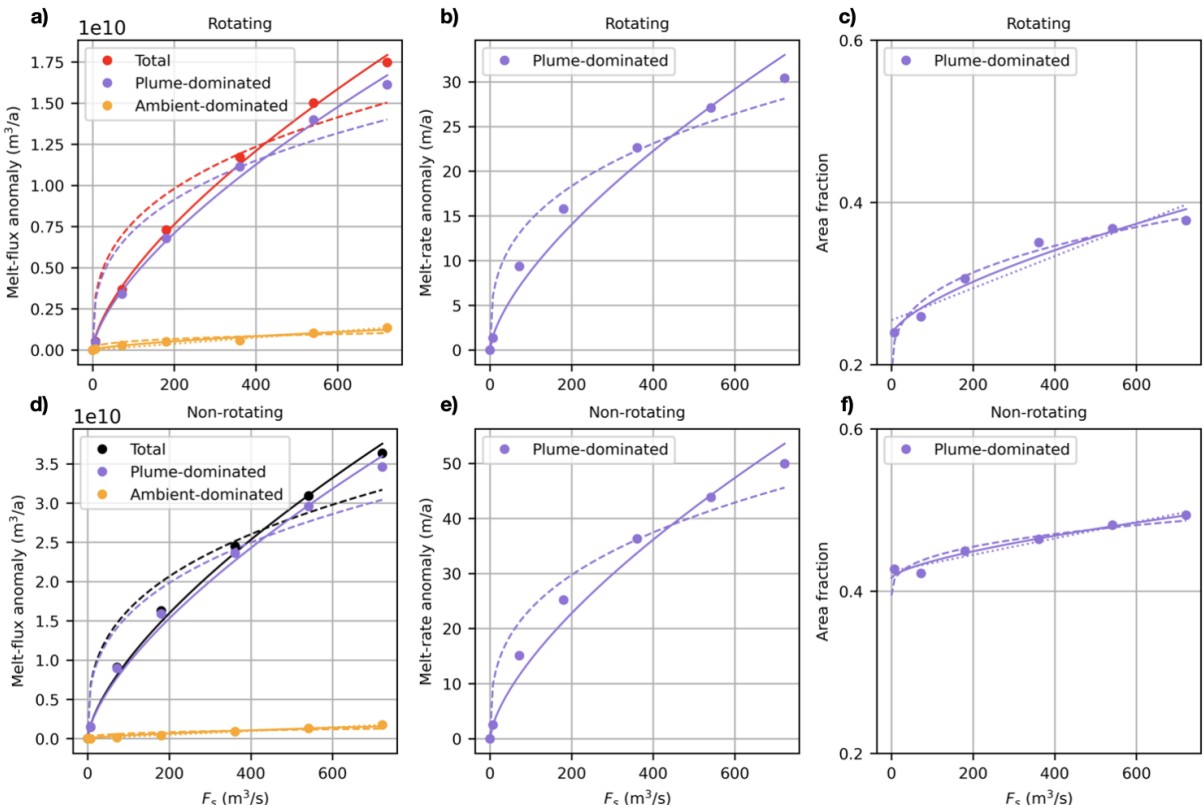

**Figure A4.** Same as Figure 9 but for distributed discharge.

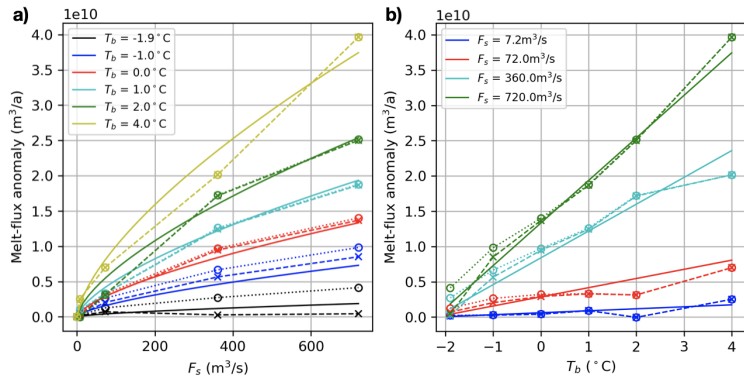

**Figure A5.** a) The relationship between the total melt-flux anomaly ($\Delta \dot{M}$) and distributed subglacial discharge ($F_s$) for different $T_b$ values. b) The relationship between $\Delta \dot{M}$ and $T_b$ for different $F_s$ values. In both panels, the circles connected by dotted lines are modeled anomalies calculated from interfacial melt rate, while the crosses connected by dashed lines are modeled anomalies calculated from total melt rate; the sum of interfacial melt rate and freezing rate due to frazil formation. The solid line is the fit from Equation A1.

*Author contributions.* IV conceptualized the research, coded the freshwater discharge flux feature into MPAS-Ocean and E3SM, conducted the simulations, analyzed the results, made the plots, and wrote the original manuscript. XAD and IV discussed and shaped the research along the way. XAD, DC, and JW provided software support. XAD, CBB, DC, AH, MH, and SFP contributed to the review and editing process. MH, SFP, and IV contributed funding and acquired resources.

*Competing interests.* The contact author has declared that none of the authors has any competing interests.

*Acknowledgements.* Support for this work was provided through the Scientific Discovery through Advanced Computing (SciDAC) program funded by the US Department of Energy (DOE), Office of Science, Advanced Scientific Computing Research and Biological and Environmental Research Programs. IV also received funding from the Laboratory Directed Research and Development program of Los Alamos National Laboratory under project number 20220812PRD4. This research used resources provided by the Los Alamos National Laboratory Institutional Computing Program, which is supported by the U.S. Department of Energy National Nuclear Security Administration under Contract No. 89233218CNA000001. We gratefully acknowledge the computing resources provided on Blues, a high-performance computing cluster operated by the Laboratory Computing Resource Center at Argonne National Laboratory. We thank two anonymous reviewers for their valuable feedback.

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
