# Peer review of "Subglacial discharge effects on basal melting of a rotating, idealized ice shelf"

_EGUsphere, 2024_

## Author Response (AR1)

REVIEWER 1

The authors conducted model experiments for an idealized ice shelf with subglacial discharge for rotating and non-rotating ice shelves. Authors show different sensitivity to subglacial discharge for rotating and non-rotating ice shelves. They also show how earth rotation and subglacial discharge location can impact cavity circulation and the melting of ice shelves. The results are well presented and key conclusions are supported by the model analyses provided in the manuscript. I suggest moderate revision.

Moderate comment

(1) Practically, all ice shelves are influenced by rotation. The meaning for considering non-rotating ice shelves, I assume, is small ice shelves that do not feel the effect of earth rotation. It would be nice to have a paragraph describing the length scale that separates rotating and non-rotating ice shelves. Maybe a discussion involving non-dimensionalized parameters may be helpful.

*The non-dimensional number that separates out rotating and non-rotating dynamics is the ratio of fjord/ice-shelf width (W) and first-mode baroclinic deformation radius (Rd). However, most Greenland's fjords live in the intermediate space where W/Rd is close to 1 and Jackson et al., 2018 nicely shows that 3D (meaning rotation is important) dynamics are still dominating this regime. This is now mentioned in this paper.*

(2) There is no discussion about vertical resolution and coordination. Is the behavior of ice shelf meltwater strongly impacted by the choice of vertical resolutions? Do you see a similar response even if you double (or half) the vertical resolutions? How sensitive is your overall conclusion to the choice of vertical resolutions?

*We have added information about the vertical mixing scheme and explained the effect of those choices as understood from previous studies. Also, we have added information about the effect and purpose of the Losch layer in MPAS-Ocean, which acts as a simplified boundary layer of spatially uniform thickness and is also used to achieve melt rate convergence with increasing vertical resolution (Gwyther et al., 2020).*

*Experiments with doubling vertical resolution indicate that the scaling relationships reported on in the paper are unaffected by resolution. The example below shows results for the base case with shelf bottom temperature of 1 degree C with rotation for varying amounts of subglacial discharge. Red is 36 layers, black 72 layers, dots connected by dotted lines are simulation results. The solid lines are discharge to power two thirds fit and the dashed lines discharge to power one third fit. The right panel shows the same but on a log log plot.*

[Figure]

Minor comment

Lines 103-105: It would be nice if authors could explain somewhere (a Table maybe) in the manuscript what the observed (or estimated) rates of subglacial discharges are for different ice shelves (that helps us understand, for example, figures 2 and 6 better).

*In this idealized study we have sampled a broad range of observed temperatures and modeled subglacial discharge values, but this is not supposed to represent any specific ice shelf. We have added a paragraph for that with a reference. Subglacial discharge rate estimates are almost exclusively based on models; there are no Antarctic-wide observational estimates.*

Lines 163-169: This text is a bit difficult to follow. Please consider adding arrows to help us where you are talking about in Figures 7e1 and e2.

*We have clarified the text.*

Citation: https://doi.org/10.5194/egusphere-2024-2297-RC1

REVIEWER 2

This manuscript presents results of idealised modeling experiments on the impact of subglacial discharge on basal melting of ice shelves. The authors vary the magnitude and distribution of subglacial discharge and find a complex set of behaviours. They also explicitly show the difference that rotation makes, which I feel is important because the temptation may be to ignore this when designing parameterisations due to the complexity introduced. They nicely contrast the results to the 'Greenlandic' vertical calving front case.

Since there is increasing interest in the role of freshwater discharge on basal melting of ice shelves, but relatively little research has been undertaken to date, I think this paper is timely and important and useful. I also find it to be generally scientifically sound but have some remaining questions and suggestions for improving the presentation. I detail these below in categories of significant comments and minor comments.

Significant comments

Vertical mixing. In general, the rate of entrainment of ambient water into a rising plume is a strong control on its dynamics – higher entrainment means the plume warms up but slows down quickly, while lower entrainment means a faster but generally colder plume. You touch on this in the paragraph beginning on L133. In your model, since the plume is travelling mostly horizontally, the mixing between plume and ambient cavity occurs in the vertical. But I saw very little discussion of vertical mixing in the methods or results. Some details that would be good to include and discuss:
(i) what is the typical vertical resolution of the model near the ice base?,

*The following text is now included in the manuscript: "Beneath ice shelves the coordinate follows the ice draft and at the seafloor layers are dropped when they intersect the bathymetry. As a result the thickness of the top vertical layer, in contact with the ice-shelf base, varies smoothly from thinnest near the grounding line (~0.75 m), thickening towards the ice-shelf front (~15 m) and thickest in the open ocean (~ 21 m)."*

[Figure]

(ii) what is the vertical mixing scheme in the model (for both momentum and tracers)?,

*For vertical mixing, we use constant vertical diffusivity - there are 2 values associated with that, one for stable, and the other for unstable case. Parameterizations and coefficients in this study follow the cited ISOMIP+ design protocol (coefficients are in Table 4 in Asay-Davis et al., 2016). This is now explicitly stated in the revised paper.*

(iii) does this vertical mixing scheme have a strong bearing on the results?,

*Another vertical scheme we have available is KPP. We have run some simulations with KPP instead of constant vertical diffusivities and there was little difference on the result. However, we note that KPP involves multiple parameters, and exploration of the sensitivity with respect to these is outside of the scope of the paper. Anyway, KPP is calibrated based on observations in the tropical ocean and is therefore not designed for ice-shelf cavities, so it is unclear whether the added complexity can be expected to bring added realism.*

(iv) can you say any more about the impact of the choices of vertically-averaging over cells within 10m of the ice (L87) and distributing the meltwater into the ocean based on an exponential profile (L88) – I imagine these could influence the results?

*The Losch layer introduces additional mixing apart from constant vertical diffusivities. It effectively creates a well mixed boundary layer of spatially uniform thickness that is used by the basal melt-rate parameterization. While in z-coordinate models it is introduced to avoid sharp meltrate changes (staircases) that would result from large differences in partial cell volumes (and therefore heat capacities), in MPAS-Ocean we introduced it to achieve melt rate convergence with increasing vertical resolution (Gwyther et al., 2020). Similarly, we assume that the vertical diffusion of meltwater may be poorly approximated by the vertically-constant diffusivity used for stable stratification, therefore meltwater distribution over a vertical length scale is introduced. It simulates meltwater distribution away from the ice base by enhanced vertical diffusion in the boundary layer. The crudeness and simplicity of our boundary layer representation is a result of lack of wider physical understanding of boundary layer processes and is typical for realistic, global model configurations.*
*The choice of 10 m thickness of the Losch layer comes with a corresponding choice of heat transfer coefficients, which are tuned to a) observations - in realistic global model configurations, or b) to target melt rates - in idealized simulations for which the truth is unknown (e.g. Asay-Davis et al., 2016). A different Losch layer thickness would require new tuning, so it is questionable how meaningful it is to test sensitivity to that parameter in our simulations whose scope lies elsewhere.*

Presentation. I found the manuscript quite hard to follow because within the first short results section (L126-149) you had already referred to 5 different results figures, each of which has a significant amount of information. As the results go on, there is a lot of jumping back and forth

between figures, often when only a small detail from the figure is relevant. Having first referenced Fig. 2 at the very start of the results (L127), Fig. 2 panels b and c are then not discussed until the very end of the results (L215), so that when I was on L127 and reading the caption to Fig. 2, I had to go looking for what Eq. 1 was.

*We have followed the suggestion and moved the original Fig. 2. The figure was split and moved to the appendix.*

I also found it a bit confusing to keep switching between 'melt rate summary figures' (Figs. 2, 6, 9) and 'simulation output details figures' (Figs. 3-5, 7, 8). At some point I found it easier to look through the figures on their own rather than trying to read the text and go to the relevant figure. I am not exactly sure how, but it would be great if there was a way of presenting this information in a more linear fashion that was easier for the reader to follow. Some ideas: could you discuss and present the simulation output details figures first, then culminate with the melt rate summary figures which provide the main conclusions of the paper? At least bringing figures 2, 6 and 9 together would make sense I think.

*Thank you for the constructive thoughts and suggestions. We split and moved the original figure 2 to appendix. The original figure 9 has been expanded by additional analysis and split into 2, one for channelized and the other for distributed discharge. We have removed the mention of meltrate-discharge scalings from the Results section so the Results are now structured as 1) the effect of discharge distribution and location on meltrates and 2) scaling relationships. However, a 'melt rate summary figure' (originally fig 6 now fig 5) is still needed relatively early in the first section, because it provides quantification of the melt-rate differences with discharge location.*

Overall importance of subglacial discharge to ice shelf melt rates and parameterisation of ice shelf melt rates. Fig. 3, left hand column, suggests that the influence of subglacial discharge is quite limited to the region close to where it is being discharged. If this is true for all reasonable values of subglacial discharge then although this was mentioned in the text, I feel that more could be made of this in the discussion and potentially the abstract. In particular in the discussion, because it feels very relevant to how we might parameterise the influence of subglacial discharge on ice shelf basal melt rates. Given the effect appears to be very localised, is a parameterisation for the area-average influence (such as Eq. 1) particularly useful?

*We weren't quite thinking of Equation 1 as a parameterization that would be directly applied in an ice-sheet model or similar. More so, we are using it to explore and understand scaling relationships between subglacial discharge and melt rate, similar to the previous two-dimensional and non-rotating three-dimensional studies. It can be used to indicate at which volume of subglacial runoff the melt rate change will be significant and that information can be used differently depending on the audience. For example, ocean modelers may choose to implement subglacial discharge into their model if the melt-rate increase is expected to be significant from the perspective of freshwater input into the southern ocean and its effects on continental shelf processes, or omit it otherwise.*

*We have now explicitly stated in the text that the melt rate change is most pronounced near the discharge location, same as in the previous non-rotating studies.*

Won't Eq. 1 be sensitive to how big the ice shelf is while also not capturing the spatial variability?

*The coefficients yes, but the scaling no. Already when averaging within 10 km from the source (at the deepest part of the grounding line, the ice shelf is 40 km wide), the melt rate scaling follows discharge to two thirds power and it stays that way if we increase distance as far as averaging over the whole ice shelf. As the coefficients of the Equation 1 do change depending on area used for averaging we have removed their values from the manuscript.*
*The scaling relationship is not designed to capture the spatial variability here, or in any of the previous two-dimensional numerical studies.*

Does it make more sense to produce a version of Eq. 1 for some region close to the discharge location?

*Close to the discharge region, before rotation becomes important, the scaling from Jenkins 2011 applies, as indicated in old Figure 9, so we do not focus on that, as it is already well established.*

Minor comments

L84 – what is the horizontal viscosity scheme? Can you add a little more detail?

*We use constant horizontal Laplacian viscosity as per the ISOMIP+ protocol. This choice was made to be small enough not to kill eddies but still be numerically stable. (values of viscosities in Table 4 in Asay-Davis et al., 2016)*

L104 – could you add what specific values of Fs you used?

*Added in a later paragraph.*

L116-124 – the set of experiments where you vary where to apply Fs in the vertical. I didn't see these mentioned in the results or discussion, but saw them plotted on Fig. 6c, where it seems this doesn't make much difference. I feel this is quite a technical, model-specific thing that has limited relevance to the dynamics of ice shelves, so perhaps move to an appendix or supporting information (and discuss the results of what it showed)? This additional complexity would help to streamline the paper.

*Thank you for pointing that out, we have followed your suggestion.*

L150 – "figures 3, 7, 4, 8 and 5" – it's not important but having these numbers out of order doesn't help with the reader being overwhelmed with swapping back and forth between figures.

*Changed.*

L223 (Eq. 1) – should k have 10^(-5) instead of 10^5? Also the units of k look correct for n=1/3 rather than 2/3. I presume this should give melt rates in m/s? If so that is probably worth stating since people will be used to thinking in m/yr.

*The values of the coefficients were removed as per major point 3 above.*

L241 – nice paragraph and discussion. The other part that is potentially relevant and could be discussed here is the influence of plume rise height. The higher the plume rises, the more ice-ocean area over which it increases the friction velocity. And since the plume will rise higher with higher discharge, this can contribute to a higher exponent n when we look at melt rates averaged over a fixed, larger area like an ice shelf. I don't know to what extent this is relevant for your results?

*We have now added additional analysis showing how the plume area in contact with the ice-shelf base scales with discharge differently for the rotating and non-rotating cases and how that impacts the observed scaling relationship for the total mean meltrate anomalies.*

Localised vs area-averaged melt rates (see also significant comment 3). I really liked Fig. 9, where you restricted to the area close to discharge outlets. I feel that the distinction between area-averaged and localised melt is perhaps not given enough attention in the manuscript. The discussion focuses a lot on the area-averaged melt rates, but for the reasons you give at the end of the discussion it may be that melt rates very close to the grounding line are particularly important. Is it worth bringing up there your results from Fig. 9 that suggest a closer to 1/3 power law dependence when close to the grounding line? And also mentioning this in the abstract?

*We didn't focus so much attention to the area close to the grounding line and the discharge region, as that is where rotation is not yet as important, and therefore the scaling from Jenkins 2011 applies, as indicated in old Figure 9, so that is already known. Similarly, the melt rate change being most pronounced near the discharge location is known from the previous non-rotating studies, so we gave it less attention here.*

Citation: https://doi.org/10.5194/egusphere-2024-2297-RC2